# Tiny Piezoelectric Multi-Layered Actuators with Application in a Compact Camera Module—Design, Fabrication, Assembling and Testing Issues

**DOI:** 10.3390/mi13122126

**Published:** 2022-12-01

**Authors:** Chao-Ping Lee, Mi-Ching Tsai, Yiin-Kuen Fuh

**Affiliations:** 1Department of Mechanical Engineering, National Central University, Taoyuan City 32001, Taiwan; 2Department of Mechanical Engineering, National Cheng-Kung University, Tainan City 70101, Taiwan

**Keywords:** piezoelectric, lead zirconate titanate (PZT), multi-layered actuators

## Abstract

Piezoelectric actuators with multi-layer structures have largely gained attention from academic and industry experts. This is due to its distinctive advantages of fast response time, huge generative force and the inherent good planar electromechanical coupling factor, as well as other mechanical qualities. Typically, lead zirconate titanate (PZT) is one of the most represented piezoelectric ceramic materials that have been used for multi-layer piezoelectric actuators. Piezoelectric multi-layered actuators (PMLAs) were developed vigorously in the past decades due to the emergence of portable devices, such as smartphones with a highly compact camera module (CCM) and an image stabilizer (IS). This study reviewed the progress made in the field of PMLA applications, with a particular focus on the miniaturized dimensions and associated generated output force, speed and maximum output power requirement for various loads. Several commercial attempts, such as Helimorph, Lobster and the two-degrees-of-freedom ultrasonic motor (USM), were investigated. The proposed simple bimorph and multi-layer bimorph USMs experimentally showed thrust as high as 3.08 N and 2.57 N with good free speed and structural thicknesses of 0.7 and 0.6 mm, respectively. When compared with the other 14 reported linear USMs, they ranked as the top 1 and 2 in terms of the thrust-to-volume ratio. The proposed design shows great potential for cellphone camera module application, especially in moving sensor image stabilization. This study also provided outlooks for future developments for piezoelectric materials, configurations, fabrication and applications.

## 1. Introduction

The thin-plate piezoelectric structure is easily manufactured and used. Due to its small volume and limited power, it is usually used for driving smaller devices but is not good for applications requiring large power. The use of multi-layer piezoelectric actuators to replace the previous bulky piezoelectric ceramics has become an important development direction to reduce the driving voltage, especially in the application of portable products that require greater output.

Yao [1] et al. proposed a longitudinal-bending rotary USM with a diameter of only 3 mm. Its structure is a Langevin transducer with a longitudinal-bending coupling mechanism, but a multi-layer piezoelectric ceramic is used to replace the traditional bulk piezoelectric ceramics, which allows the motor to be driven at 10 V. Aoyagi [2] et al. proposed a USM using multi-layer piezoelectric ceramics. This design uses two stacked piezoelectric actuators to regularly generate linear motion and rotational motion by rubbing the sides of the rotor.

Multi-degree-of-freedom motors were also researched. Toyama [3] presented a USM with three rotational degrees of freedom based on the stator of the Sashida motor, which provided an interesting idea for the development of multi-degree-of-freedom USMs. However, using multiple stators increases the complexity of the structure and falls out of line with cost considerations.

Regarding a thin-plate structure, Aoyagi [4] et al. designed a multi-degree-of-freedom USM with different driving modes by exciting the divided electrodes. In particular, the driving frequencies of different modes are sent to different electrodes at the same time, thereby enhancing the desired amplitude by combining different modes. Takemura [5] et al. proposed a thin-plate type USM with different driving phases to generate multi-degree-of-freedom motion. He attached piezoelectric ceramic plates with different divided electrodes to an amplitude amplification mechanism made of copper metal. Motion with multiple degrees of freedom is created by driving different phases to different electrodes. Otokawa [6] et al. developed an array plate structure. Taking advantage of the fact that specific positions on a single plate have different directions of motion under different driving modes, four plate stators are arranged in an array. Different modes are excited for different stators to produce movements with different degrees of freedom.

In recent years, researchers have paid more attention to linear motors. The linear ultrasonic motor performs 4–5 times faster in terms of focusing speed than the electromagnetic stepping motors installed in a similar lens. As with the miniature rotary ultrasonic motors, which are installed at the side of the lens and used with linear guide mechanisms, the advantage of using the linear ultrasonic motor is that a direct drive without transmission mechanism and backlash can be achieved [7]. In all the research papers published between 2015 and 2020 about USMs, only 5% considered multi-DOF linear USMs, such as planar USMs [8]. For the reported planar USM, it is a challenge to fit them into the cellphone camera module, such as a bolt-clamped stator [9]. Other than that, there are other interesting innovations that linear USMs provide. Some vibration modes were introduced, such as an electrode cross-connected type B1 mode for the standing mode and double B1 mode for the traveling mode [10,11]. Other examples include in-plane expending and bending modes [12] and face-diagonal-bending mode [13], which induce in-plane vibration modes that will drive a slider using the friction material on the side.

These designs were commercialized and some are still in the market, such as Squiggle [14], Miniswys [15], TULA [16] or Butterfly [17]. TULA reported their progress on a multi-layer stator that reduces the driving voltage, but there is no news on the development of a multi-DOF USM by TULA or any other brand. A piezoelectric single crystal was also reported to be used for a micromotor [18,19,20]. Due to the difficulty in fabrication, the production cost becomes a major issue, and thus, a single crystal is not easy to promote. Izuhara [21] et al. introduced a metallic ring motor with eight small PZT rectangular plates that combine longitudinal and bending modes to generate linear motion and can keep track. The performance may not be so good, but the design is quite creative.

Piezoelectric ultrasonic motors may be the most prominent micromotor because of their high torque density and simple components. In fact, linear ultrasonic motors dominate in millimeter-scale applications, such as camera modules [22]. There are many review papers about USMs that reported the development trend of miniaturization and low driving voltage. A low operating voltage for deploying in a portable device and a smaller-sized motor structure to improve compactness has been researchers’ main focus in recent years [23]. The rapid development of multilayer piezoelectric ceramics and micromachining technology has great prospects for ultrasonic motors to reach large amplitudes with low-voltage drives. This would, in turn, have an important influence on the integration of ultrasonic motors, lightening the overall weight and enabling a real sense of no electromagnetic interference [24].

In this paper, a two-degree-of-freedom USM based on a bimorph actuator is presented. In order to understand the benefits of a multi-layer structure, a conventional bonded, metal-shim-type simple bimorph USM, and a novel multi-layer bimorph USM was fabricated and tested simultaneously. The driving mechanism of the two planar moving directions was on the opposite side of the bimorph stator to avoid interference while moving. The most unique advantage of this design is that the thickness was 0.7 mm for the simple bimorph stator and 0.6 mm for the multi-layer stator. The test result showed that both the thrust-to-volume ratio and the free-speed-to-volume ratio were excellent compared with the performance of other reported USMs. The thin-plate structure is very promising for a mobile phone camera module, especially for a moving-sensor-type image stabilization module, which needs a compact multi-DOF actuator.

## 2. Piezoelectric Multi-Layered Actuators (PMLAs) as Previously Applied to Compact Camera Modules (CCMs)

### 2.1. Helimorph Actuator Based on Superhelix Structures with Bimorph and Multi-Layered Configurations

It has always been challenging to directly use the deformation of the piezoelectric actuator inverse piezoelectric effect to achieve displacement targets. The displacement is usually too small for compact portable products. Amplifying the displacement in a limited space through the design of the structure is an issue that must be considered. The Helimorph actuator launched by One Limited in 2002 is one of the best. The bimorph actuator is spirally turned in a ceramic green tap, rolled into an open ring structure with a fixed inner diameter and sintered (Figure 1a), which maximizes the effective size of the bimorph actuator. The annular structure can also be perfectly combined with the lens module (Figure 1b).

Theoretical designs of piezoelectric ceramic helical and spiral actuators have existed for many years. The main reason these devices have not achieved widespread recognition and use is the perceived difficulty of processing such forms [25]. The former Sunnytec Electronics manufactured a Helimorph actuator licensed by One Limited that started with a simple two-layer bimorph structure. After sintering, the two-layer Helimorph has a total thickness of 320 µm and a 9.5 mm annular outer diameter. Under a driving voltage of −80 V to +80 V, the no-load free displacement can exceed 725 µm.

Laminating the piezoelectric ceramic structure is the most viable method to reduce its driving voltage. Helimorph, whose main market was mobile digital cameras, could not ignore this important design trend. By maintaining the same driving voltage, the displacement and thrust of the actuator can be increased, thereby improving the possibility of increasing the size and number of lenses. Figure 1d shows the cross-section of the multi-layer Helimorph; it can be seen that the total thickness is maintained at 320 µm but is divided into eight layers, each with an average of 40 µm, at the same voltage of −80 V to +80 V under driving conditions. The no-load free displacement of the Helimorph actuator tip can be increased to 1450 µm, which greatly improves the load capacity of the Helimorph actuator and fully met the needs of the emerging mobile phone digital camera autofocus lens at the time of its design.

The complex process, structure and fragility of the ceramic structure limit the market development of thin and long piezoelectric actuators, such as Helimorph actuators. After all, the product has not entered the market. Simplifying the process and structure to improve the robustness of the product is the future.

### 2.2. Lobster Actuator for a Compact Camera Module (CCM)

With the demand for optical zooming, the requirements for actuator displacement often require more than 2 mm. Because of the low stroke (less than 0.20 mm), pure actuator units, such as bimorphs and multilayers, cannot be used, and the current applicable designs are grouped into two categories: impact drive and ultrasonic motor [26]. Sunnytec Electronics developed a multilayer impact-drive-type motor [27]. The motor shown in Figure 2 is named “Lobster” due to its clamping part design.

Lobster is composed of a multi-layer piezoelectric actuator, a metal clamp, a rod slider and a metal base. Different from the smooth-impact drive mechanism (SIDM) of Konica-Minolta [28] and the tiny ultrasonic linear actuator (TULA) [29] developed by Piezo Tech that fixed the rod on the piezoelectric element as a stator to push the sliding clamp lens on the clamping part, Lobster uses the design of a fixed rod and clamping part slider. The metal clamping part is fixed on the multi-layer piezoelectric actuator. When the piezoelectric actuator vibrates, the metal clamping part vibrates up and down to push the lens fixed on the rod slider as an SIDM. The advantage of this is that the rod that determines the longest moving distance is not fixed on the lens module base structure, but on the moving lens, thereby reducing the thickness of the mechanism structure, which provides more flexible design options for the telescopic lenses module type that require a thinner installation size.

In addition, the two fingers of the lobster clamping part have different thicknesses. From the simulation results in Figure 3, it can be found that the thinner finger on the left will be distorted when it vibrates up and down. With proper design, this mode of movement can increase the efficiency of an SIDM motor. In the sliding phase of the movement, the gap between the rod and the clamp is increased to reduce friction, while in the frictional push phase, the corresponding friction force is still maintained.

The experimental results show that under a driving voltage of 5.30 V, the maximum moving distance of each step of the upward movement is approximately 23 nm, the effective moving distance after deducting the rebound is approximately 17 nm and the maximum moving speed is approximately 4 mm/s. Calculated at a frequency of 89.74 kHz, the average moving speed is approximately 1.53 mm/s. Under a driving voltage of 4.20 V, the maximum moving distance of each step of the upward movement is approximately 18 nm, the effective moving distance after deducting the rebound is approximately 10 nm and the maximum moving speed is approximately 3 mm/s. Calculated at a frequency of 91.74 kHz, the average moving speed is approximately 0.92 mm/s. The measurement results are shown in Figure 4b,c.

Lobster can be used for AF or optical zooming. Figure 5 demonstrates one lobster USM assembled into a 9.50 mm × 9.50 mm AF module housing. It shows the possibility to move a lens for an AF application and the very good potential for optical zooming since the movement can be over 2 mm or even longer.

In addition, Figure 6 demonstrates the concept of a two-axis stabilizer module integrated with Lobster USM as in Figure 6a the two-axis stage design and Figure 6b the fabricated prototype image stabilizer with dimensions of 17 mm × 12 mm × 4 mm. The displacement limits of the two axes are limited by the plastic mechanism. The module is driven by a function generator and a power amplifier and it is measured using a Doppler laser vibrometer. The maximum displacement depends on the length of the zirconia rod and the movement can be greater than 500 µm with a good response, which satisfies the goal of an optical IS.

## 3. Two-Degrees-of-Freedom USMs—Design, Fabrication, Assembling and Testing Issues

### 3.1. Design Concept

A bimorph is a commonly used piezoelectric actuator structure, which is composed of two flat piezoelectric ceramics. After different polarization procedures, parallel or series bimorphs can be formed. Because the driving voltage of the parallel bimorph is relatively low, it is more suitable for the purpose of this study. In order to properly lead out the electrodes between the two ceramic plates, a layer of metal shim is usually used between the two piezoelectric ceramic plates.

In addition, through an analysis of the literature, it was found that in order to cause the bimorph-type USM to produce different driving directions, it is often necessary to give different phases. The bimorph can be divided into two electrode areas to give different phase signals so that the bimorph is bent as desired. If the electrode division is further extended to another dimension to form a four-block electrode structure, a bimorph stator with two degrees of freedom can be obtained.

Furthermore, to reduce the driving voltage without changing the external dimensions of the piezoelectric element, the use of a multi-layer structure is a widely used method.

The type of piezoelectric materials (PZT) used for the bimorph actuators was PZT-5. The basic parameters were as follow: d_33_ = 500 × 10^−12^ m/V, K_p_ = 0.62, Q_m_ = 65, ε_33_ = 3400 and tanδ = 1.8%.

Figure 7a shows the size and structure design of the simple bimorph stator. Here, two piezoelectric plates with a size of 10 mm × 10 mm × 0.30 mm were attached to both sides of a 10 mm × 10 mm × 0.10 mm copper shim plate, and the piezoelectric plate had a structure divided by four electrodes to drive the movement in two degrees of freedom. The copper shim was designed with a protruding part for the electrode connection.

The design of the multi-layer bimorph stator is shown in Figure 7b. The size was 10 mm × 10 mm × 0.60 mm to compare the performance under the same piezoelectric ceramic volume. In terms of the electrode design, there were also four divided electrodes to drive the motion in two degrees of freedom.

When making a bimorph stator, in order to make the two sides of the piezoelectric stator drive in different directions, a friction tip design as shown in Figure 8a is considered. The friction tip is attached at different symmetrical positions on the up and down sides, and the friction tip is cut flush with one edge of the ceramic, 3 mm away from the adjacent vertical side. The friction tip on the upper side is responsible for the movement along the *X*-axis, while the friction tip on the lower side is responsible for the movement along the *Y*-axis. The size of the friction tip is shown in Figure 8b and the material is copper.

### 3.2. Fabrication and Assembly

The simple bimorph stator is formed by combining two single-layer ceramic plates with piezoelectric properties, and the bonding method is generally via gluing. Through the polarization direction and the design of the driving electrode, the expansion and contraction directions of the two piezoelectric materials are different, causing bending deformation. In order to increase the rigidity of the component, avoid breakage and facilitate the extraction of electrodes, a layer of metal is also pasted between the two layers of ceramics.

Because of the flexibility of the process, the multi-layer bimorph stator can be directly designed to meet the electrode requirements in the ceramic green body during the screen-printing and stacking process to form two regions with different amounts of expansion and contraction to achieve bending deformation. After the ceramic is completed, a bimorph with a complete structure is formed, which does not require a bonding process, but rather provides a convenient solution for the cost and reliability of mass production. The driving voltage can be easily reduced by using a thinner single layer. Figure 9 shows the internal multi-layer structure of the fabricated element, with dimensions of 10 mm × 10 mm and a total thickness of 0.62 mm after sintering. The thickness of the single layer is 56 µm, the total number of layers is 10 and each side has five layers. There are 28 µm non-piezoelectric protective layers on the top and bottom of the element, which is convenient for making external electrodes in the future. The ceramic is sintered densely, and the thickness of the inner electrode is uniform and continuous. It is a good multi-layer piezoelectric element, which can be used for subsequent stator production. The actual stator is shown in Figure 10, in which Figure 10a is a simple bimorph stator and Figure 10b is a multi-layer bimorph stator.

After completing the fabrication of the two types of bimorph stator, the resonant frequency and displacement amplitude were measured next. An Agilent HP4294A impedance analyzer was used to measure the resonant frequency. At the same time, the bimorph was driven with a function generator, and the displacement was measured for each frequency using a Doppler laser interferometer to compare the relationship between the simulation and the measured results. Since the friction tip was designed as the output position of the stator vibration, the position of the friction tip was also used as the measurement point during measurement.

Figure 11a shows the frequency–phase–impedance diagram of the simple bimorph stator. The resonant frequencies of the three fundamental bending modes were 25.21 kHz, 57.20 kHz and 93.25 kHz, respectively. Among the three modes, the resonant waveform of the first mode was the most complete. Then, a function generator was used to drive the simple bimorph stator with a voltage of 10 V at different frequencies, and a laser interferometer was used to measure the displacement amplitudes of the *X*-axis and *Z*-axis. The results are shown in Figure 11b; the first mode also had higher displacement amplitudes on the *X*-axis and *Z*-axis, where the displacement amplitude of the *X*-axis was 1.26 µm, the displacement amplitude of the *Z*-axis reached 2.69 µm and the ratio of the two-axis displacement amplitude was 2.14.

Figure 11c shows the frequency–phase–impedance diagram of the multi-layer stator. The resonant frequencies of the three fundamental modes were 20.71 kHz, 47.65 kHz and 82.13 kHz, respectively. Among the three modes, the resonant waveform of the first mode was also relatively complete, and the multi-layer bimorph stator was driven by a function generator with a voltage of 10 V at different frequencies. A laser interferometer was used to drive the simple bimorph stator. The displacement amplitude of the *X*-axis and *Z*-axis were measured and the results are shown in Figure 11d. The figure also shows that the first mode had higher displacement amplitudes on both the *X*-axis and the *Z*-axis, where the displacement amplitude of the *X*-axis was 1.14 µm, the displacement amplitude of the *Z*-axis was 3.88 µm and the ratio of the two-axis displacement amplitude was 3.41.

After completing the simulation, the impedance analysis measurement and the displacement amplitude measurement using the laser interferometer, the simulation and actual measurement of the simple piezoelectric stator and the multi-layer piezoelectric stator were compared and the results are summarized in Table 1 and Table 2. There were several important results found that are worth discussing:
(a)First of all, it can be noted that the first three fundamental bending modes were found in the finite element method simulation, impedance analyzer measurement and displacement amplitude measurement, which showed that the simulation and the actual measurement were related.(b)The measured results showed that both the simple bimorph stator and the multi-layer bimorph stator had a large displacement amplitude in the first mode; therefore, subsequent experiments can be performed in this mode.(c)The error between the simulation and the actual measurement expanded with the increase in the modal frequency, in particular, the errors of the third mode reached 14.96% (simple type) and 11.71% (multi-layer type), respectively. The main reason was that the friction tip was not added in the simulation, but the friction tip was added in the actual measurement. Although this error existed, it had no effect on the chosen first mode, and thus, related experiments can still be carried out.(d)In the displacement amplitude measurement of the laser interferometer, it was found that the resonant frequencies of the *X*-axis and *Z*-axis of each mode were sometimes slightly different, which may have been due to the measurement error or the selection of the measurement position.(e)The *Z*-axis displacement ratio of the multi-layer bimorph to the simple bimorph for the first, second and third modes were 14.42, 10.20 and 46.00, respectively. These ratios seem too high since the thickness-per-layer ratio of a simple bimorph to multi-layer bimorph was only 5.36. This meant that the electric field of the multi-layer bimorph should also be 5.36 times the simple bimorph under the same driving voltage. The reason may have come from the difference in the dimensions and structure of the two designs. The simple bimorph had a 0.1 mm copper layer and was also 0.1 mm thicker than the multi-layer bimorph; therefore, the simple bimorph was more structurally robust. Meanwhile, the *X*-axis displacement ratios were 9.02, 7.38 and 4.03, respectively.

### 3.3. Testing and Performance Evaluation

The setup of the USM performance measurement system is shown in Figure 12. During the experiment, according to the frequency required by the vibration mode, two sets of sine waves with a phase difference of 90 degrees were generated by the function generator, and the two sets of power amplifiers were raised to their respective required voltages. The boosted signal was sent to the two groups of driving electrodes of the piezoelectric stator so that the stator could vibrate properly, thereby pushing the slider.

In the design of the slider, a linear guideway was used to reduce the frictional resistance of the movement, and an alumina plate was attached to the slider to increase the friction between the stator and the slider so that the output energy of the stator could be transmitted to the slider as much as possible. The contact friction between the stator and the slider has a great influence on a USM. If the friction is too small, the driving force will not be fully transmitted and the efficiency of the slider movement will be reduced; if the friction is too large, the slide will be obstructed and the maximum efficiency will not be achieved. Therefore, careful adjustment of the pre-load has a great impact on the performance of a USM. While applying the pre-load, a thrust meter is used for measurement.

In terms of speed measurement, a simple distance and time measurement method was used. By measuring the time for moving a known distance, the average moving speed of the slider can be calculated. The speed measured here was free speed under no load. In the terms of thrust measurement, a load cell was used to measure the thrust, and the thrust measured was the maximum thrust under the condition of zero movement speed.

The driving frequency will also have an impact on the performance of the USM. In order to understand the optimal efficiency of the USM that was made, the optimal frequency for driving the USM was tested when conducting the experiments. After the simple bimorph stator was installed on the test platform, the resonant frequency measured was 26.30 kHz.

Since the USM uses frictional motion as the method for energy transmission and the pre-load has a major impact on the friction force, different pre-loads were first tested to find out the range of pre-loads required to drive the USM. Subsequently, the free speed and the maximum thrust were measured, and the influence of these characteristics was observed under different driving voltages.

The free speed vs. pre-load and the maximum thrust vs. pre-load curves of the simple bimorph USM under different driving voltages are shown in Figure 13a,b, respectively. As for the free speed, under different voltages, the free speed increases relatively with the increase of the pre-load when the USM starts to move. Taking the driving voltage of 60 V as an example, the USM started to drive when the pre-load was 1.10 N and the free speed was 24.05 mm/s. As the pre-load increased, the free speed also increased, reaching a maximum value of 35.82 mm/s at 1.90 N, and then decreased as the pre-load increased until it could not be driven.

The entire pre-load–free speed curve will move up as the driving voltage increases. In the meantime, with different driving voltages, the optimal pre-load value also changes. As the driving voltage increases, the optimized pre-load increases. When the driving voltage was 120 V, the optimal pre-load value was 2.50 N and the free speed reached 87.75 mm/s.

As far as maximum thrust is concerned, the pre-load–thrust curve exhibits the same behavior as free speed. When the driving voltage was 120 V and the pre-load was 2.50 N, the maximum thrust obtained was 3.08 N.

Generally, the control of the pre-load while assembling a USM is almost inflexible, especially for a miniature USM. Therefore, it is necessary to select a specific pre-load as the assembly parameter. From the characteristic curve of the pre-load, it can be found that when the pre-load was 2.50 N, the USM had better performance, and thus, a pre-load of 2.50 N was chosen for the follow-up experiment.

After deciding on the pre-load of 2.50 N, the free speed and maximum thrust could be obtained under each driving voltage. However, since the pre-load was fixed, the displayed free speed and maximum thrust were not necessarily the same as the maximum values of the optimal pre-loads under the specific driving voltage. The free speed vs. driving voltage curves and the maximum thrust vs. driving voltage curves at the optimal pre-loads compared with the 2.50 N pre-load are shown in Figure 14a,b, respectively. It can be found from the figure that if the pre-load was selected to be 2.50 N, the simple bimorph USM did not show the best performance due to the difference between the pre-load and the maximum free speed of each voltage, especially when the drive voltage was low.

This section discusses the performance of the output power and efficiency. First, the free speed and maximum thrust were marked under different driving voltages on the speed–thrust diagram and it was assumed that there was a linear relationship between speed and thrust. Figure 15a shows the speed vs. thrust diagram at different driving voltages. The output power is the product of speed and thrust and dividing the output power by the input power gives the efficiency of the USM. The relationship between thrust and output power is shown in Figure 15b. It can be found from the figure that the maximum output power reached 64.67 mW under a driving voltage of 120 V. Further observing the efficiency situation, it was found that with the increase of input voltage, the efficiency also improved. Under the driving voltage of 120 V, the maximum efficiency was 1.80%. The relationship between the thrust and output power is shown in Figure 15c.

When looking at the efficiency relationship, it was observed that there was a phase difference between the input voltage and current, that is, the power factor was lower than 1. It can also be found that in most cases, the power factor was lower than 0.50 or even as low as 0.20. After correcting for the power factor, it was found that the maximum efficiency was increased to 4.03%. The efficiency vs. thrust curve after the power factor correction is shown in Figure 15d.

The same analysis process was used in the performance of the multi-layer bimorph USM to perform the pre-load and driving voltage analysis, output power and efficiency analysis, as well as correct the power factor caused by the phase difference between the input voltage and current. The resonant frequency used for the multi-layer bimorph USM was 21.00 kHz.

The free speed vs. pre-load and the maximum thrust vs. pre-load curves of the multi-layer bimorph USM under different driving voltages are shown in Figure 16a,b, respectively. When the pre-load was 1.90 N, the overall performance of the motor was better, and thus, 1.90 N was chosen as the pre-load. Under the condition of a 24 V driving voltage and a 1.90 N pre-load, the free speed was 74.23 mm/s and the maximum thrust was 2.57 N, which were slightly lower than for the simple bimorph USM under the same driving electric field.

The free speed vs. driving voltage curves and the maximum thrust vs. driving voltage curves under the optimal pre-load and 2.50 N pre-load were compared, as shown in Figure 17a,b. It can be found from the figure that the optimal pre-load of the multi-layer USM had little influence on the input voltage, where except for the 12 V input voltage condition, the optimal pre-load was 1.90 N.

The speed, output power and efficiency vs. thrust curves of the multi-layer bimorph USM are shown in Figure 18. Under the condition of a 24 V driving voltage, the maximum output power was 45.75 mW, which was lower than the 64.67 mW of the simple bimorph USM with the same driving electric field, but the efficiency was higher than that of the simple type, which were 2.20% without power factor correction and 11.63% with power factor correction, which were 1.22 times and 2.77 times the respective measured values for the simple bimorph USM of 1.80% and 4.03%, respectively. This result showed that the multi-layer bimorph USM had significantly better efficiency.

Figure 19a,b shows the performance comparison of the free speed, maximum thrust and output power between the two-layer simple bimorph USM and the ten-layer multi-layer bimorph USM under different driving voltages. It was found that the driving voltage of the multi-layer bimorph USM was significantly lower than that of the simple bimorph USM, but it could maintain a satisfactory overall output performance.

Looking at the efficiency comparison of the two types of USM, the efficiency of the multi-layer type was significantly higher than that of the simple type. At the same time, a comparison with the same electric field was made, which also showed a better efficiency performance of the multi-layer bimorph USM, regardless of whether the power factor was corrected or not. So far, it can be deduced that the multi-layer bimorph USM had the advantage of being applied under a low driving voltage and was suitable for use in portable products. The efficiency comparison is shown in Figure 19c,d.

The multi-layer bimorph USMs had efficiencies higher than the simple bimorph because of the difference in structure. The multi-layer bimorph stator was a monolithic ceramic that was fabricated using a co-fired process to obtain a uniform structure. The inner electrode was the major factor that decreased the electromechanical coupling coefficient, which meant that the efficiency of the piezo actuator was affected negatively. Since the inner electrode was very thin, the effect was not as significant. The simple bimorph stator in this study had two single-layer PZT plates bonded with a 0.1 mm copper shim. There were two major factors that affected the efficiency. First, the copper shim occupied more than 15% mass of the stator, which was definitely a heavy load on the stator. Second, regarding the adhesive for bonding, compared with the inner electrode, the adhesive affected the efficiency even more. Finding the right adhesive and applying it correctly are always key issues for bimorph actuators.

Both the multi-layer bimorph USM and the simple bimorph USM were inefficient. The reason for this might have been the soft PZT-5 material. PZT-5 is good for building multi-layer actuators due to its high piezoelectric constants (dij), wide working frequency band and commercial availability. However, the large material loss and temperature rise also make the USM inefficient [14]. Further study may consider using a hard PZT material.

Performance comparisons of the USMs for this study and 14 other reported linear USMs are listed in Table 3. The criteria for selection were their size, performance or applications, including in a digital camera, which were indicated in their respective reports. The table shows that the volume of the most selected USMs were between 13.5 mm^3^ and 450 mm^3^. The volume of the simple bimorph USM and multi-layer bimorph USM were 70 mm^3^ and 60 mm^3^, respectively, which were not very small and have room to improve. The free speeds of the USMs were between 5 mm/s and 310 mm/s, while simple and multi-layer bimorph USM free speeds were 87.85 mm/s and 74.34 mm/s respectively. The requirement of the working speed for a cellphone camera autofocusing function is around 20 mm/s with an approximately 20 mN loading. In this case, higher is better. Therefore, the free speed of the USMs in this study was good enough for the proposed application. The thrust provides the best advantage of the proposed USM. The thrust was 3.08 N at a 120 V driving voltage for the simple bimorph USM and 2.57 N at 24 V for the multi-layer bimorph USM, which are ranked number 2 and number 4 of the 16 selected USMs, respectively. If the sizes are factored-in, the two proposed USMs will perform even better and will become the top 2 of the 16 USMs. The relationship between the free speed and the thrust force of the selected USMs is also shown in Figure 20.

## 4. Conclusions

(a) Previously attempted PMLAs, such as Helimorph and Lobster, were briefly reviewed and some practical issues regarding their commercialization, such as the manufacturing yield (fabrication, assembly, testing, etc.), were also considered for practical adoption into consumer electronics. To avoid fabrication and assembly issues, precise control of the thickness, surface finishing and curvature of the PZT actuators should be meticulously monitored to follow the design and implementation specifications of the proposed CCM structures.

(b) The novel PMLA-based Helimorph structures utilize multi-layers for high displacements at low voltages, which represents a three-dimensional super helix structure, but have serious fabrication issues. Compared with the conventional two-dimensional planar multi-layer processing, the crucial difference is the necessity to physically bend the laminated structures. The performance characteristics rely heavily on several fabrication steps, such as the layer thickness, the number of layers, the diameter of the helix, the flexibility of the tape, the alignment of the electrode design, and the configuration during binder burnout and sintering.

(c) The Lobster actuator represents one of the piezoelectric actuators that has been developed for several decades due to the impact-drive-type mechanism, which was largely classified as a smooth impact drive mechanism (SIDM). An SIDM-type actuator of Lobster was introduced by integrating the PMLA and pre-defined shape of metallic structures. The aim of various types (that have camera phone potentials) is to include both an optical zoom mechanism (a stroke of more than 2 mm) and the auto-focusing function (only a 0.20 mm motion of the lens), which can be realized using an impulse piezoelectric motor.

(d) The proposed two-degrees-of-freedom piezoelectric multi-layer bimorph USM was designed, simulated, fabricated and measured, confirming that the first bending mode was suitable for driving the bimorph stator, and the simulation and measurement results were consistently matched such that both the simple type and the multi-layer piezoelectric bimorph USM could be driven under an electric field of more than 200 V/mm.

(e) Experimenting with an electric field of 400 V/mm, the simple bimorph USM had a maximum thrust of 3.08 N, a maximum speed of 87.75 mm/s and a maximum efficiency of 4.20%. The multi-layer bimorph USM stator had a maximum thrust of 2.58 N, a maximum speed of 74.23 mm/s and a maximum efficiency of 11.63%. In addition, the pre-load had a great influence on the performance of the USM and needed to be selected carefully. The experimental results showed that the optimal pre-loads required by the simple and multi-layer bimorph USMs were 2.50 N and 1.90 N.

(f) The use of multi-layer bimorph stators to make USMs has a significantly lower driving voltage and higher efficiency, which has great potential for development. The implementation of a dual-axis USM shows that it can be driven at voltages above 12 V and has the potential to be used in portable products.

(g) A comparison between the proposed bimorph USMs and 14 other reported linear USMs was analyzed in this report. The proposed USMs showed extremely good thrust performance with good free speed given their compact size. It provides a lot of room for further size reduction for a better and more compact design.

(h) During the experiment, it was found that the proposed USMs became hot after some time, which reduced their performance. However, using a hard PZT material will be the next step and should improve the performance.

## Figures and Tables

**Figure 1 micromachines-13-02126-f001:**
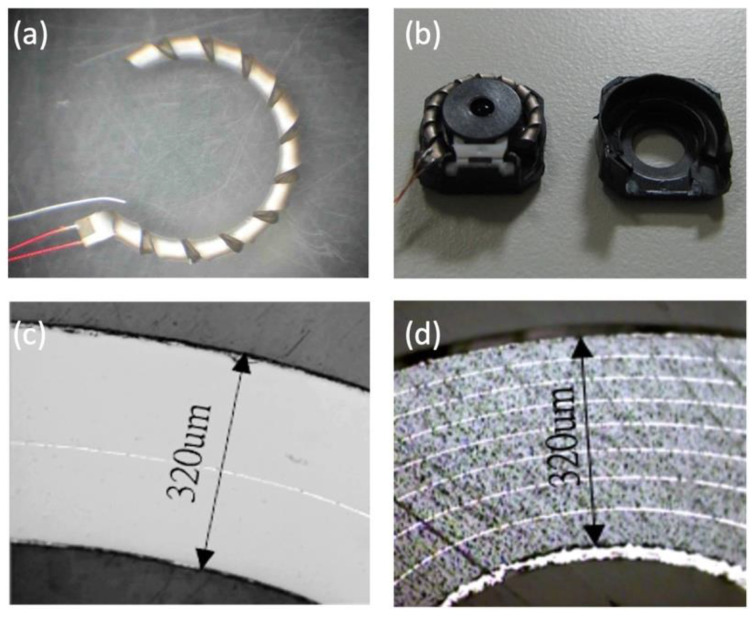
(**a**) Helimorph actuator and (**b**) camera module manufactured by the former Sunnytec Electronics licensed by the former One Limited. (**c**) Cross-section of a 2-layer Helimorph. (**d**) Cross section of an 8-layer Helimorph.

**Figure 2 micromachines-13-02126-f002:**
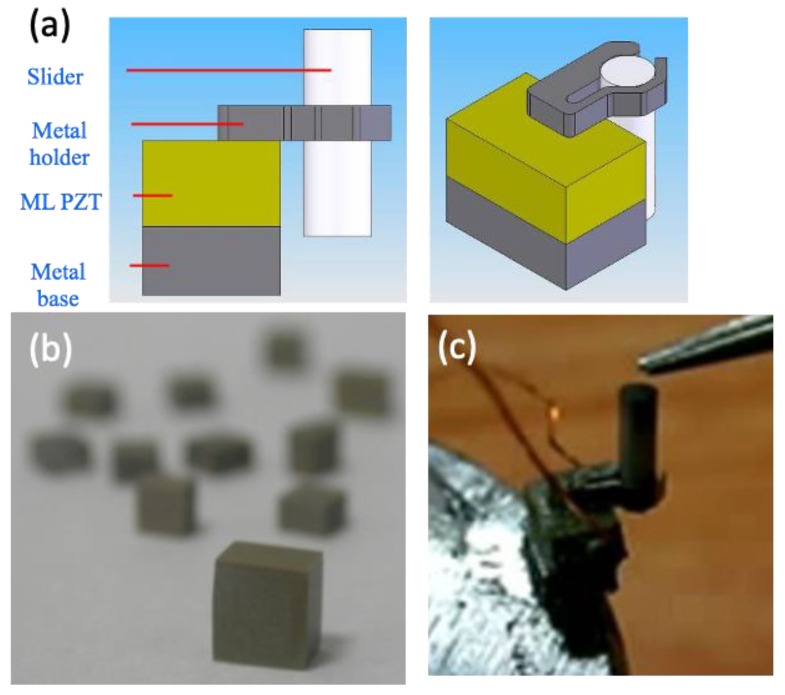
(**a**) Lobster USM design. (**b**) ML PZT for Lobster USM. (**c**) Lobster USM.

**Figure 3 micromachines-13-02126-f003:**
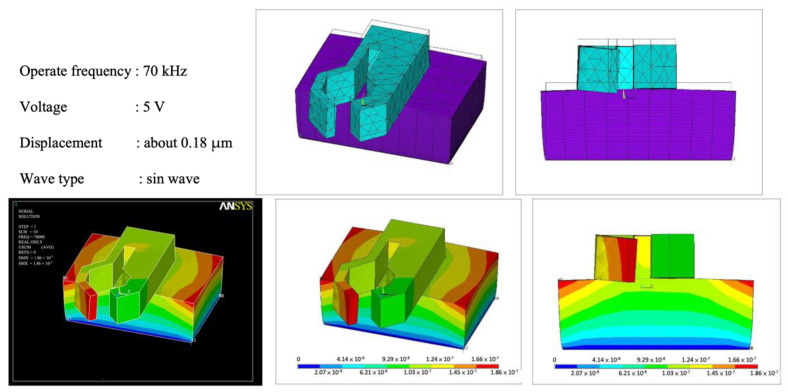
Lobster USM simulation result.

**Figure 4 micromachines-13-02126-f004:**
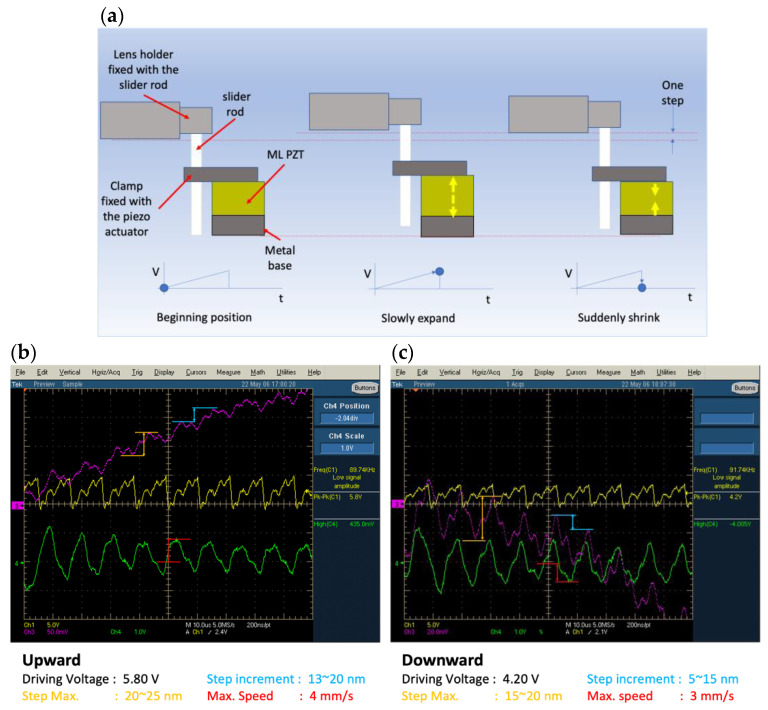
(**a**) SIDM movement of the Lobster USM and Lobster USM moving performance. (**b**) Upward movement at a 5.80 V driving voltage. (**c**) Downward movement at a 4.20 V driving voltage.

**Figure 5 micromachines-13-02126-f005:**
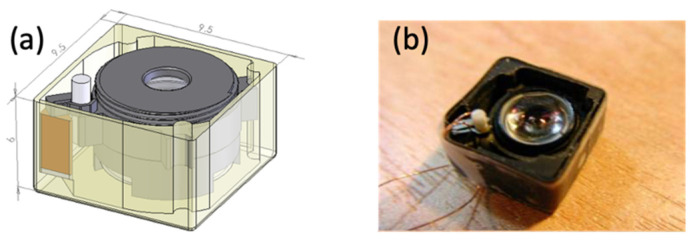
AF module with a Lobster USM. (**a**) The design. (**b**) The prototype with dimensions of 9.50 mm × 9.50 mm × 6 mm.

**Figure 6 micromachines-13-02126-f006:**
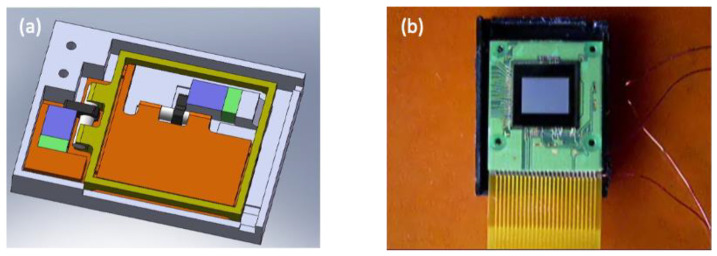
Two-axis stabilizer module with a Lobster USM. (**a**) The two-axis stage design. (**b**) The prototype image stabilizer with dimensions of 17 mm × 12 mm × 4 mm.

**Figure 7 micromachines-13-02126-f007:**
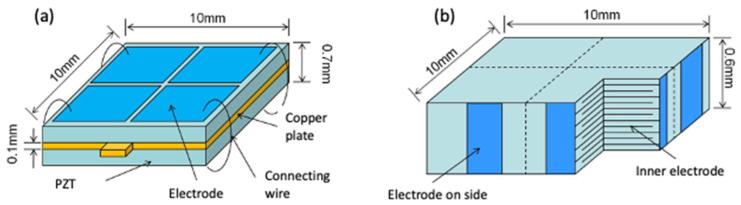
Schematic design of (**a**) bimorph and (**b**) multi-layer piezoelectric actuators.

**Figure 8 micromachines-13-02126-f008:**
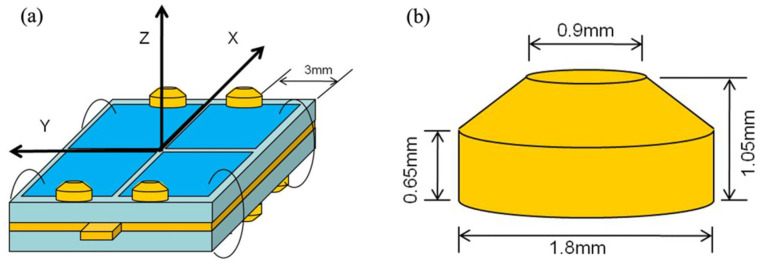
(**a**) Bimorph stator and (**b**) the friction tip design.

**Figure 9 micromachines-13-02126-f009:**
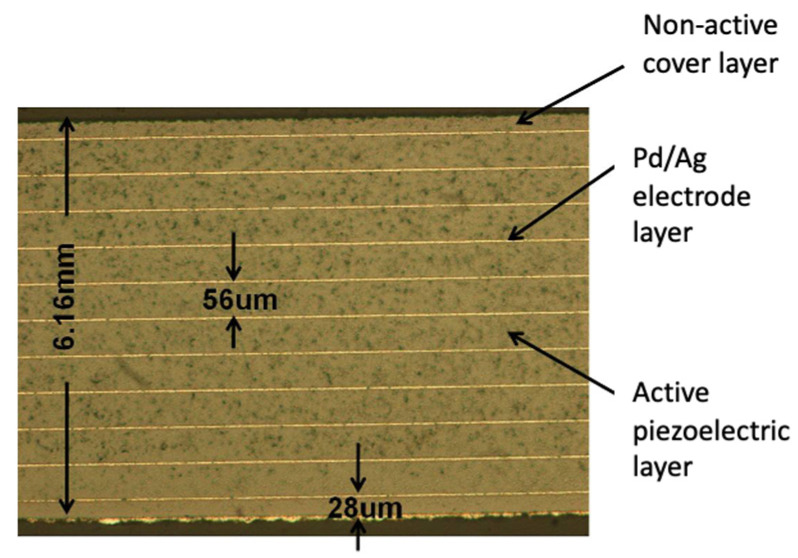
Ten-layer piezoelectric bimorph stator with 56 µm per layer.

**Figure 10 micromachines-13-02126-f010:**
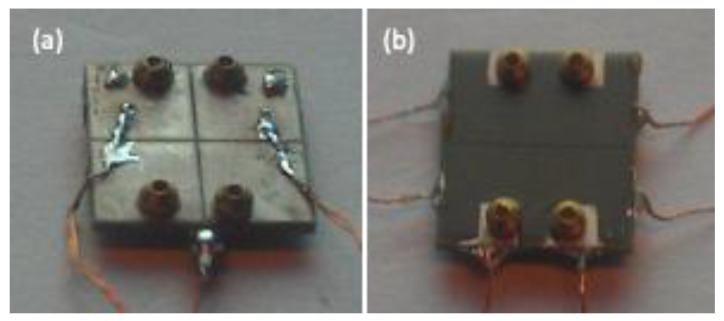
Fabricated stators. (**a**) Simple bimorph and (**b**) multi-layer bimorph.

**Figure 11 micromachines-13-02126-f011:**
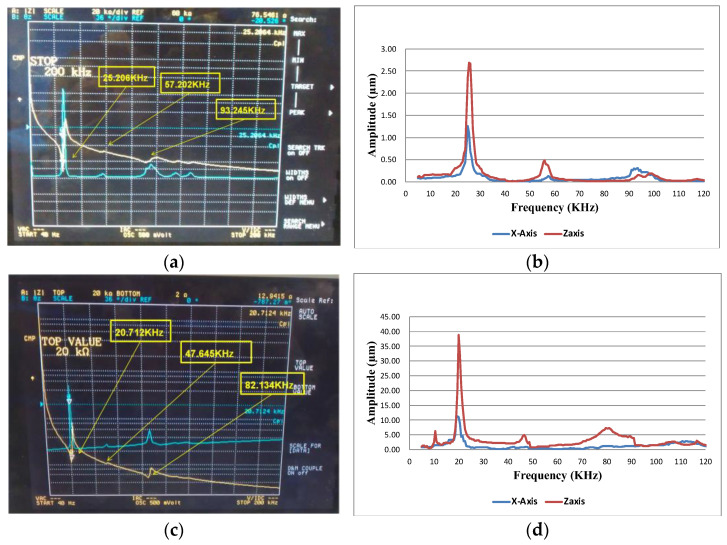
(**a**) The frequency, impedance and phase analysis; (**b**) the displacement–frequency response diagram of the multi-layer bimorph stator; (**c**) the frequency, impedance and phase analysis; and (**d**) the displacement–frequency response diagram of the simple bimorph stator.

**Figure 12 micromachines-13-02126-f012:**
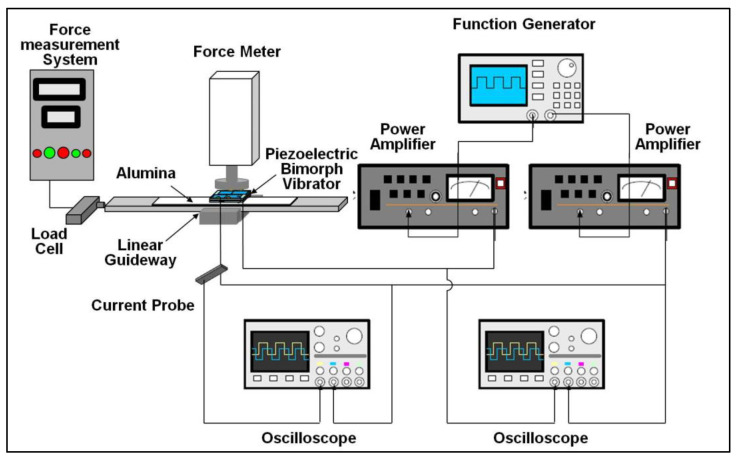
USM performance measurement system.

**Figure 13 micromachines-13-02126-f013:**
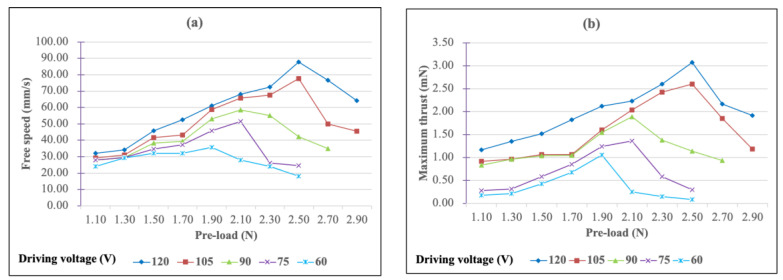
(**a**) Free speed and (**b**) maximum thrust figures as a function of pre-load for the simple bimorph USM under different driving voltages.

**Figure 14 micromachines-13-02126-f014:**
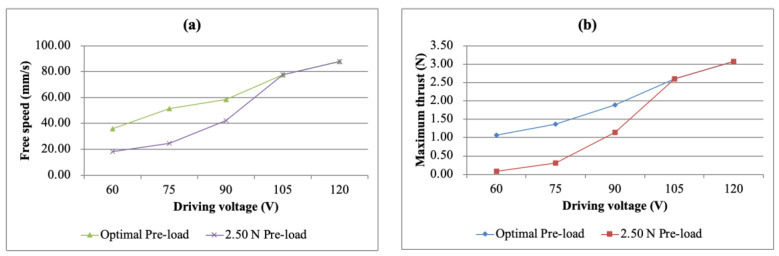
(**a**) Free speed and (**b**) maximum thrust as a function of the driving voltage for the simple bimorph USM under different pre-loads.

**Figure 15 micromachines-13-02126-f015:**
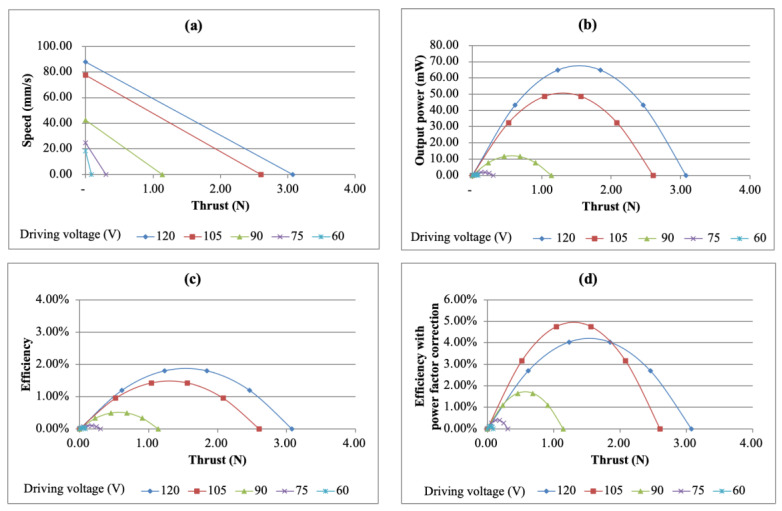
Performance of the simple bimorph USM under different driving voltages. (**a**) Speed vs. thrust curve, (**b**) output power vs. thrust curve, (**c**) efficiency vs. thrust curve and (**d**) efficiency vs. thrust curve with power factor correction.

**Figure 16 micromachines-13-02126-f016:**
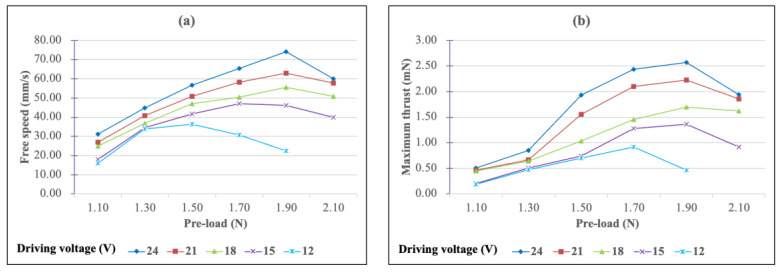
(**a**) Free speed and (**b**) maximum thrust figures as a function of the pre-load for the multi-layer bimorph USM under different driving voltages.

**Figure 17 micromachines-13-02126-f017:**
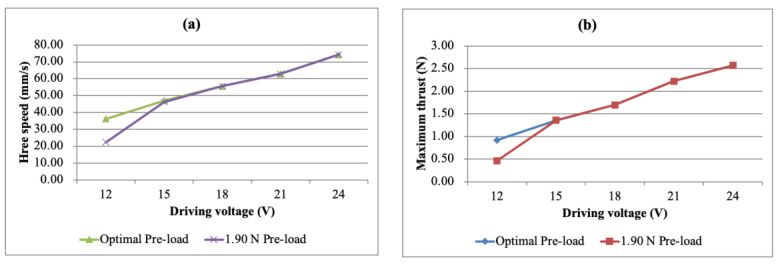
(**a**) Free speed and (**b**) maximum thrust figures as a function of the driving voltage for the multi-layer bimorph USM under different pre-loads.

**Figure 18 micromachines-13-02126-f018:**
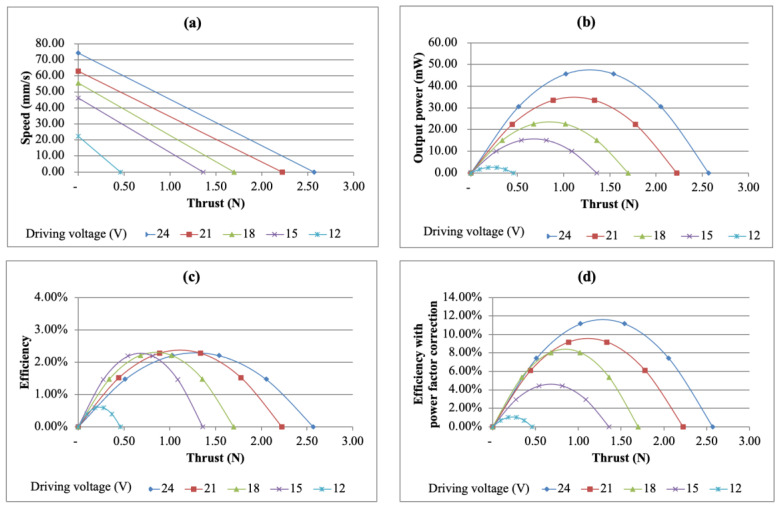
Performance of the multi-layer bimorph USM under different driving voltages. (**a**) Speed vs. thrust curve, (**b**) output power vs. thrust curve, (**c**) efficiency vs. thrust curve and (**d**) efficiency vs. thrust curve with a power factor correction.

**Figure 19 micromachines-13-02126-f019:**
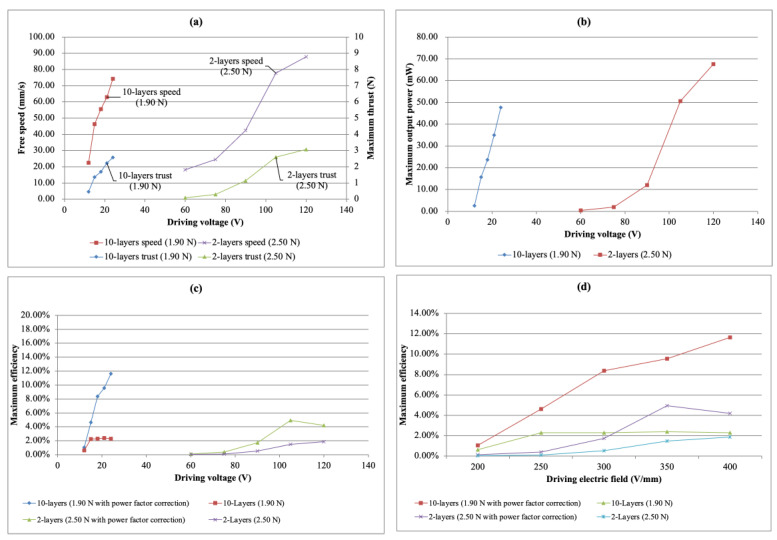
The performance comparison of the simple bimorph USM and the multi-layer bimorph USM. (**a**) Driving voltage vs. free speed and driving voltage vs. maximum thrust, (**b**) maximum output power vs. driving voltage, (**c**) maximum efficiency vs. driving voltage and (**d**) maximum efficiency vs. driving electric field.

**Figure 20 micromachines-13-02126-f020:**
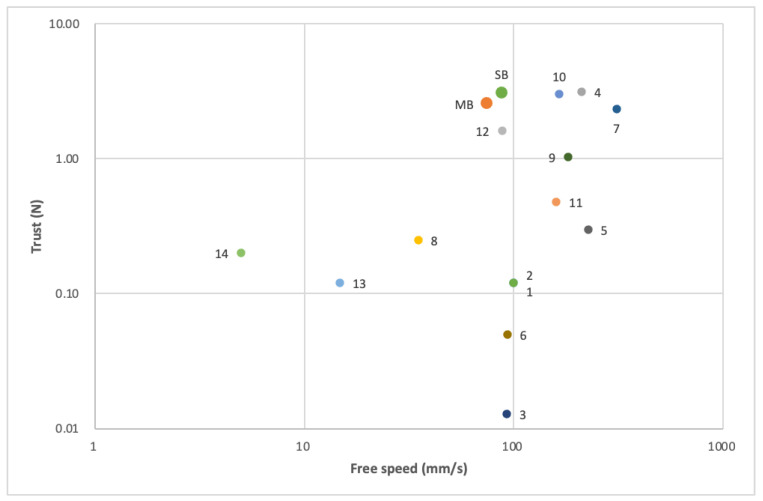
Comparison of the USMs: the relationship between the free speed and the thrust force.

**Table 1 micromachines-13-02126-t001:** Comparison between the simulation and measurement of the simple bimorph stator.

Items	First Mode	Second Mode	Third Mode
Simulation frequency (kHz)	24.60	59.61	109.64
Impedance analyzer measurement frequency (kHz)	25.21	57.20	93.25
*X*-axis laser interferometer measurement frequency (kHz)	25.00	57.50	93.50
*X*-axis displacement (µm@10V)	1.26	0.13	0.31
*Z*-axis laser interferometer measurement frequency (kHz)	25.00	56.00	93.50
*Z*-axis displacement (µm@10V)	2.69	0.49	0.16
Difference between impedance analyzer measurement frequency and simulation frequency	2.47%	4.03%	14.96%
Measurement frequency difference between impedance analyzer and laser interferometer (*X*-axis)	0.82%	0.52%	0.27%
Measurement frequency difference between impedance analyzer and laser interferometer (*Z*-axis)	0.82%	2.10%	0.27%

**Table 2 micromachines-13-02126-t002:** Simulation and measurement comparison of the multi-layer bimorph stator.

Items	First Mode	Second Mode	Third Mode
Simulation frequency (kHz)	20.71	49.87	93.03
Impedance analyzer measurement frequency (kHz)	20.71	47.65	82.13
*X*-axis laser interferometer measurement frequency (kHz)	19.50	47.20	81.00
*X*-axis displacement (µm@10V)	11.37	0.96	1.25
*Z*-axis laser interferometer measurement frequency (kHz)	20.00	46.70	81.00
*Z*-axis displacement (µm@10V)	38.79	5.00	7.36
Difference between impedance analyzer measurement frequency and simulation frequency	0.02%	4.47%	11.71%
Measurement frequency difference between impedance analyzer and laser interferometer (*X*-axis)	5.85%	0.93%	1.38%
Measurement frequency difference between impedance analyzer and laser interferometer (*Z*-axis)	3.44%	1.92%	1.38%

**Table 3 micromachines-13-02126-t003:** Comparisons of the USMs.

USM Code	Reference	Year	Type	DOF	Dimension mmL × mmW × mmt	Volume (mm^3^)	Voltage of the Max. Free Speed (V)	Free Speed (mm/s)	Voltage of the Max. Thrust (V)	Thrust (N)	Free Speed /Volume {(mm/s)/mm^3^}	Thrust /Volume (N/mm^3^)
SB	This study	2022	Simple Bimorph	2	10 × 10 × 0.7	70.00	120.00	87.75	120.00	3.08	1253.57	0.0440
MB	This study	2022	Multi-Layer bimorph	2	10 × 10 × 0.6	60.00	24.00	74.23	24.00	2.57	1237.17	0.0428
1	[19]	2022	PZT	1	6.7 × 4.2 × 0.5	14.07	22.62	100.00	22.62	0.12	7107.32	0.0085
2	[19]	2022	Single crystal	1	6.7 × 4.2 × 0.5	14.07	28.28	100.00	28.28	0.12	7107.32	0.0085
3	[21]	2021	PZT	1	4.5 × 4.5 × 0.9	18.23	100.00	92.80	100.00	0.01	5091.91	0.0007
4	[9]	2016	PZT	2	63 × 10 × 10	6300.00	400.00	211.30	400.00	3.15	33.54	0.0005
5	[11]	2015	PZT, Traveling wave	1	5 × 2 × 2	20.00	70.00	227.00	80.00	0.30	11,350.00	0.0150
6	[11]	2015	PZT, Standing wave	1	5 × 2 × 2	20.00	80.00	93.00	80.00	0.05	4650.00	0.0025
7	[12]	2015	PZT	1	15 × 15 × 2	450.00	150.00	310.00	150.00	2.35	688.89	0.0052
8	[20]	2014	Single crystal	2	9 × 2 × 2	36.00	80.00	35.00	80.00	0.25	972.22	0.0069
9	[18]	2013	Single crystal	1	9.6 × 9.6 × 2.5	230.40	50.00	182.50	50.00	1.03	792.10	0.0045
10	[13]	2013	PZT	1	15 × 15 × 2	450.00	150.00	165.00	150.00	3.00	366.67	0.0067
11	[10]	2012	PZT	1	12 × 4 × 4	192.00	40.00	160.00	40.00	0.48	833.33	0.0025
12	[17]	2011	PZT	1	9 × 8 × 1	72.00		88.00		1.62	1222.22	0.0225
13	[16]	2009	PZT	1	11 × 4 × 2.5	110.00	12.00	14.80	12.00	0.12	134.55	0.0011
14	[14]	2006	PZT	1	6 × 1.5 × 1.5	13.50	40.00	5.00	40.00	0.20	370.37	0.0148

## Data Availability

Not applicable.

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
