# Peer review of "Tiny Piezoelectric Multi-Layered Actuators with Application in a Compact Camera Module—Design, Fabrication, Assembling and Testing Issues"

_micromachines, 2022, doi:10.3390/mi13122126_

Round 1

Reviewer 1 Report

I have the following comments for the authors to help them improve their manuscript:

1. Many language problems can be found in the manuscript. The writing should be largely improved.

2. What's the uniqueness and innovation of the study? The authors should clearly clarify them in the introduction section. 

3. The authors suggest to comparing the experiment results with the reported results.

4. The abstract and conclusion section need to rewrite. 

Author Response

Dear Reviewer,

Thank you for your input to make this manuscript better. Please find the response in the attached file.

Best Regards,

Chao-Ping Lee

Reviewer 2 Report

Dear Authors,

My review points are listed in the attachment. 

Kind Regards

Author Response

(The authors gave the same response as above.)

Round 2

Reviewer 1 Report

I do not have further technical comments for the authors. However, many language and grammar problems can still be found in the manuscript. The authors should ask help from a native speaker for improving writing.  

Author Response

Dear Reviewer,

Thanks for your input to make the manuscript better. Please see my response in attached file.

Best Regards,

Chao-Ping Lee

Reviewer 2 Report

Dear Authors,

You have improved the paper according to my comments. There are still some minor format and methodological errors and mistakes. The paper can be published after minor revision. 

Kind Regards

Author Response

(The authors gave the same response as above.)
